# AN ANALYTIC FRAMEWORK FOR ROBUST TRAINING OF DIFFERENTIABLE HYPOTHESES

## ABSTRACT

The reliability of a learning model is key to the successful deployment of machine learning in various industries. Creating a robust model, particularly one unaffected by adversarial attacks, requires a comprehensive understanding of the adversarial examples phenomenon. However, it is difficult to describe the phenomenon due to the complicated nature of the problems in machine learning. Consequently, many studies investigate the phenomenon by proposing a simplified model of how adversarial examples occur and validate it by predicting some aspect of the phenomenon. While these studies cover many different characteristics of the adversarial examples, they have not reached a holistic approach to the geometric and analytic modeling of the phenomenon. Furthermore, the phenomenon have been observed in many applications of machine learning, and its effects seems to be independent of the choice of the hypothesis class. In this paper, we propose a formalization of robustness in learning theoretic terms and give a geometrical description of the phenomenon in analytic classifiers. We then utilize the proposal to devise a robust classification learning rule for differentiable hypothesis classes and showcase our proposal on synthetic and real-world data.

## 1 INTRODUCTION

The state-of-the-art machine learning models are shown to suffer from the phenomenon of adversarial examples, where a trained model is fooled to return an undesirable output on particular inputs that an adversary carefully crafts. While there is no consensus on the reasons behind the emergence of these examples, many facets of the phenomenon have been revealed. Szegedy et al. (2014) show that adversarial perturbations are not random and they generalize to other models. Goodfellow et al. (2015) indicate that linear approximations of the model around a test sample is an effective surrogate for the model in the generation of adversarial examples. Tanay & Griffin (2016) show that adversarial examples will appear in linear classifiers when the decision boundary is tilted towards the manifold of natural samples. Ilyas et al. (2019) reveal that the distribution of training samples and robustness of the trained model are related. Demontis et al. (2019) have proposed three metrics for measuring transferability between a target and a surrogate model based on the similarity of the loss landscape and the derivatives of the models. Li et al. (2020) infer that the cause of the phenomenon is probably geometrical and that statistical defects are amplifying its effects. Barati et al. (2021) find an example that shows pointwise convergence of the trained model to the optimal hypothesis is enough for the phenomenon to emerge. Shamir et al. (2021) explore the interaction of the decision boundary and the manifold of the samples in non-linear hypotheses.

However, there are some issues with the current proposals in the literature. The geometrical and computational descriptions of the phenomenon does not always agree in their predictions (Moosavi-Dezfooli et al., 2019; Akhtar et al., 2021). Even though geometrical perspectives have the advantage of being applicable to all hypothesis classes, they are not verifiable without a computational description. On the other hand, computational approaches are mostly coupled with a particular construction of a hypothesis and in turn need a geometric description to be applicable to different scenarios. The current defence methods does not appear to be very effective (Machado et al., 2021) and a need for novel ideas is felt (Bai et al., 2021).

Our aim in this paper is to devise a framework for the analysis of the adversarial examples phenomenon that

1. is decoupled from the underlying representation of the hypothesis.
2. provides the means for visualization of the phenomenon.
3. models the known characteristics of the phenomenon as much as possible.

To this end, we first describe a necessary condition for robustness that we derive from the first principles of learning theory and lay out the groundwork for the analysis of the phenomenon from the perspective of learning rules in section 2. Next, we will extend the framework with the necessary definitions so that it would be open to geometrical interpretations in section 3. Finally, we put the proposed framework to use and verify its predictions for small scale synthetic and real-world problems. We provide a summary of the framework in appendix A for the interested reader.

## 2  PRELIMINARIES

In learning theory we are interested to study how machines learn (Shalev-Shwartz & Ben-David, 2014). We will base our analysis on the framework of probably approximately correct (PAC) learning. The basic objects of study in learning theory are hypothesis classes and learning rules. We are interested in determining the necessary condition for a learning rule $\mathcal{A}$ to be robust with respect to a nonuniform learnable hypothesis class $\mathcal{H}$. We assume that the training samples are labeled by the true labeling function and that samples come from a compactly supported distribution throughout the paper.

**Definition 2.1** (perplexity). Consider a universally consistent learning rule $\mathcal{A}$ and training sets $S', S \subset \mathcal{X}$. The perplexity of $h = \mathcal{A}(S)$ with respect to $h' = \mathcal{A}(S')$ is

$$\|h' - h\|_\infty = \sup_{x \in \mathcal{X}} |h'(x) - h(x)|. \tag{1}$$

We expect that the perplexity of the output of a learning rule $h \in \mathcal{H}$ to decrease as we add more natural samples to the training set. In contrast, adding an adversarial sample would be perplexing for $h$. If we find some sample $x \in \mathcal{X}$ that does not perplex $h$ but is not correctly labeled by $h$, then we assume that $\mathcal{H}$ is agnostic to the pattern of $x$. In other words, if adversarial training cannot improve the robustness of $h$, then $h$ is as robust as it gets for hypotheses in $\mathcal{H}$.

**Definition 2.2** (robust learning rule). Consider a learning rule $\mathcal{A}$ and a sequence $\{d_n\}_{n=1}^\infty$ of random variables that almost surely converge to a random variable $\mathcal{D}$ that is uniformly distributed on $\mathcal{X}$. $\mathcal{A}$ is robust if for every $\epsilon > 0$, there exists a natural number $N$ such that for all $m, n \geq N$ we have that

$$\|\mathcal{A}(d_m) - \mathcal{A}(d_n)\|_\infty \leq \epsilon. \tag{2}$$

This definition of a robust learning rule could be interpreted as the characteristic that when $\mathcal{A}$ observes enough natural samples, adding more samples would not be consequential to the output of $\mathcal{A}$, independent from the generative process of the samples. The definition is equivalent with Cauchy's criterion for uniform convergence of $\{\mathcal{A}(d_n)\}_{n=1}^\infty$ to the optimal hypothesis $\mathcal{A}(\mathcal{D})$.

The largest hypothesis class that we consider here is $L^2(\mathcal{X})$. The main characteristic of functions in $L^2(\mathcal{X})$ is that they are square-integrable. Formally, for a function $f \in L^2(\mathcal{X})$,

$$\|f\|_{L^2(\mathcal{X})} = \left( \int_\mathcal{X} |f(x)|^2 \, dV(x) \right)^{\frac{1}{2}} < \infty. \tag{3}$$

**Theorem 2.3.** $L^2(\mathcal{X})$ *is nonuniform learnable.*

We assume that a hypothesis $h \in \mathcal{H}$ has a series or an integral representation,

$$h(x) = \sum_{i=1}^\infty a_i \varphi_i(x), \tag{4} \qquad\qquad h(x) = \int_\Omega \nu(\omega)\sigma(x;\omega) \, d\omega. \tag{5}$$

The series representation is customary in machine learning. A series representation is adequate if we have a discreet set of features $\{\varphi_i\}_{i=0}^\infty$, e.g. polynomials. We argue that an integral representation is more adequate for analysis when there is a continuum of features to choose from, e.g. a neuron in an artificial neural network (ANN). Informally, the integral representation would abstract away the complexity of finding good features by incorporating every feature possible.

**Definition 2.4** (SVC learning rule). Consider a training set $S = \{(x_n \in \mathcal{X}, t(x_n))\}_{n=1}^N$ and a hypothesis $h \in L^2(\mathcal{X})$. The support vector classifier (SVC) learner solves the following program,

$$\underset{\boldsymbol{a}}{\arg\min} \quad \frac{1}{2}\|\boldsymbol{a}\|^2 \qquad (6) \qquad \underset{\nu}{\arg\min} \quad \frac{1}{2}\|\nu\|_{L^2(\Omega)}^2 \qquad (7)$$
$$\text{subject to} \quad t_n h(x_n) \geq 1 \qquad\qquad \text{subject to} \quad t_n h(x_n) \geq 1$$

**Theorem 2.5.** *The SVC learning rule is not robust with respect to $L^2(\mathcal{X})$, and the optimal solution is,*

$$a_i = \sum_{n=1}^N \lambda_n t_n \varphi_i(x_n), \qquad (8) \qquad \nu(\omega) = \sum_{n=1}^N \lambda_n t_n \sigma(x_n; \omega), \qquad (9)$$

*in which $\{\lambda_n\}_{n=1}^N$ are the Lagrange multipliers.*

**Proposition 2.6.** *The SVC learning rule would return the labeling function $t$ as the output in the infinite sample limit unless $\mathcal{H}$ is agnostic to $t$.*

Based on proposition 2.6, one may suggest that SVC learners with respect to $L^2(\mathcal{X})$ are weak against black box attacks as well. We formalize the conditions for transfer of adversarial examples as follows.

**Definition 2.7** (transfer of adversarial examples). The adversarial examples of a surrogate $\hat{h} \in \hat{\mathcal{H}}$ would probably approximately transfer to a target $h \in \mathcal{H}$ if some $\epsilon, \epsilon', \delta \geq 0$ exist in which with probability $1 - \delta$ we have that

$$\|\hat{h} - h\|_\infty \leq \epsilon, \qquad (10)$$

$$\|\frac{\partial \hat{h}}{\partial x_i} - \frac{\partial h}{\partial x_i}\|_\infty \leq \epsilon'. \qquad (11)$$

The idea behind this definition of transfer is that not only $\hat{h}$ and $h$ probably approximate each other, they also vary in a similar manner in the neighborhood of a test sample. We introduce normal hypothesis classes to formalize the conditions for the transfer of adversarial examples between different representations of a hypothesis class.

**Definition 2.8** (normal hypothesis class). $\mathcal{H}$ is a normal hypothesis class if for every universally consistent learning rule $\mathcal{A}$ and all $\epsilon \geq 0$ and for all sequences $\{d_n\}_{n=1}^\infty \overset{\text{a.s.}}{\to} \mathcal{D}$ a number $N \in \mathbb{N}$ exists where for all $n \geq N$ we have that

$$\|\mathcal{A}(d_n) - \mathcal{A}(\mathcal{D})\|_\infty \leq \epsilon, \qquad (12)$$

for a dense subset of $\mathcal{X}$.

**Theorem 2.9.** *If $\mathcal{H}$ and the class of its derivatives $\partial\mathcal{H}$ are normal hypothesis classes, then adversarial examples of $h \in \mathcal{H}$ would transfer between different representations of $h$.*

**Theorem 2.10.** *$L^2(\mathcal{X})$ is not a normal hypothesis class.*

Even though theorem 2.3 shows that it is possible to find a robust learning rule for $L^2(\mathcal{X})$, we argue that this hypothesis class is not appropriate for the analysis of the adversarial examples phenomenon in ANNs. Since it is not a normal hypothesis class, the hypotheses in $L^2(\mathcal{X})$ are not clustered together and it does not model the transferability of the adversarial examples between ANNs with different activation functions.

In this section we were focused on describing the phenomenon and the conditions for robustness and transfer in learning theory. Next, we will extend the framework with the tools and definitions needed for a geometrical treatment of the phenomenon.

## 3 THE SPACE OF HOLOMORPHIC HYPOTHESES

We need to come up with a suitable frame and axis if we want to be able to visualize the phenomenon. In this paper, we are only concerned with visualizing the phenomenon in binary classification tasks. We are interested in visualizing the submanifold of the natural samples $\mathcal{S}$ and to study its interactions with the submanifold of the decision boundary $\mathcal{C}$.

**Definition 3.1** (complex-valued classifier). A complex-valued classifier $h : \mathcal{X} \to \mathbb{C}$ is a function $h(x) = u(x) + iv(x)$ in which the real part $\Re[h(x)] = u(x) = 0$ encodes the geometrical position of the decision boundary and the imaginary part $\Im[h(x)] = v(x) = 0$ regresses through the geometrical position of the training samples.

The complex plane $\mathbb{C}$ provides us with a frame to visualize $\mathcal{S}$ with respect to $\mathcal{C}$. The decision boundary is represented by the imaginary axis and $h$ assigns a position in the complex plane to each $x \in \mathcal{X}$. Thus, the image of every path $\gamma : [0, 1] \to \mathcal{X}$ would be a curve $h(\gamma)$ in $\mathbb{C}$. If $\gamma$ crosses $\gamma'$ in $\mathcal{X}$, $h(\gamma)$ would cross $h(\gamma')$ in $\mathbb{C}$ as well. However, in general the geometrical interactions between $h(\gamma)$ and $h(\gamma')$ does not translate to $\gamma$ and $\gamma'$. In particular distance and angles measured in $\mathbb{C}$ may not reflect the true distance and angles in $\mathcal{X}$. in order to overcome this obstacle, we will turn to a special subset of $L^2(\mathcal{X})$.

**Definition 3.2** (the Bergman space). Consider a compact and simply connected domain set $\mathcal{X} \subset \mathbb{C}^d$. The Bergman space $A^2(\mathcal{X}) \subset L^2(\mathcal{X})$ is a reproducing kernel Hilbert space defined as

$$A^2(\mathcal{X}) = \{ f \in \mathcal{O}(\mathcal{X}) \mid \Big( \int_{\mathcal{X}} |f(z)|^2 dV(z) \Big)^{\frac{1}{2}} < \infty \}. \tag{13}$$

$\mathcal{O}(\mathcal{X})$ is the space of holomorphic functions on $\mathcal{X}$. There are different equivalent ways to characterize holomorphic functions. The holomorphic functions are the solutions to the homogeneous Cauchy-Riemann equations, or its counterpart in higher dimensions $\bar{\partial}$ (del-bar) equations. Equivalently, the holomorphic functions are the functions that are complex differentiable in each dimension of $x \in \mathcal{X}$. A third characterization of $\mathcal{O}(\mathcal{X})$ is that these functions are complex analytic and have a power series representation. $\mathcal{O}(\mathcal{X})$ is also special because, unlike real analytic functions, it is closed under uniform convergence. In one complex dimension, holomorphic functions are also known as conformal maps. Conformal maps are those maps that preserve the angle between paths in $\mathcal{X}$ and their image in $\mathbb{C}$. In other words, if $\gamma$ and $\gamma'$ cross each other at a right angle, $h(\gamma)$ and $h(\gamma')$ would cross each other at a right angle as well.

However, it could be shown that holomorphic functions can only be defined over the field of complex numbers. Nevertheless, we argue that the simplicity of analysis in $A^2(\mathcal{X})$, and its unique set of properties would justify the switch to complex numbers. We emphasize that encoding data using complex numbers could be as simple as $1 + ix$ or $e^{ix}$. We will replace $x$ with $z$ to symbolize the transition from the real to the complex number system.

A domain $\mathcal{X}$ that allows for the definition of holomorphic functions is called a domain of holomorphy. Fortunately, most of the domains that we would normally face in applications are domains of holomorphy. A notable subset of domains of holomorphy are convex domains. We have provided a short summary of the topic in appendix E, with some examples on how to encode real data with complex numbers. Krantz (2001) is our main reference for the definitions and notation in function theory of several complex variables.

**Definition 3.3** (the Bergman kernel). The Bergman kernel $K_{\mathcal{X}}(z, \zeta)$ of a compact and simply connected domain $\mathcal{X}$ is the unique function with the reproducing property

$$f(z) = \int_{\mathcal{X}} f(\zeta) K_{\mathcal{X}}(z, \zeta) dV(\zeta), \quad \forall f \in A^2(\mathcal{X}). \tag{14}$$

The reproducing property of the Bergman kernel of $\mathcal{X}$ in conjunction with the fact that the optimal Bayes classifier achieves the minimum of the 0-1 loss function provides the means to define the infinite sample limit of any learning rule on $A^2(\mathcal{X})$ that is minimizing the complex 0-1 loss function,

$$\text{loss}_{\mathbb{C}}(t, z, h) = \text{loss}_{0-1}(t, z, \Re[h]) + \text{loss}_{MSE}(0, z, \Im[h]), \tag{15}$$

independently from the details of the implementation or the training process.

**Definition 3.4** (holomorphic optimal Bayes classifier). The holomorphic optimal Bayes classifier is the orthogonal projection of the optimal Bayes classifier $f_{\mathcal{D}} : \mathcal{X} \to \mathbb{R}$ into $A^2(\mathcal{X})$,

$$o_{\mathcal{D}}(z) = \int_{\mathcal{X}} f_{\mathcal{D}}(\zeta) K_{\mathcal{X}}(z, \zeta) \, dV(\zeta), \tag{16}$$

**Theorem 3.5.** *$A^2(\mathcal{X})$ and $\partial A^2(\mathcal{X})$ are normal hypothesis classes.*

Theorem 3.5 predicts that adversarial examples of a nonrobust hypothesis $h \in A^2(\mathcal{X})$ transfer between different representations of $h$.

Geometrical properties of holomorphic functions would enable us to infer the geometrical relation between $\mathcal{S}$ and $\mathcal{C}$ by studying their images in $\mathbb{C}$. This property of the holomorphic hypotheses would prove to be key to finding a robust learning rule for $A^2(\mathcal{X})$. Next, we will extend the SVC learning rule to accommodate for complex-valued hypotheses.

**Definition 3.6** (complex SVC learning rule). The complex SVC learner solves the following program,

$$
\begin{aligned}
\arg\min_{h,\xi} \quad & \frac{1}{2}\|h\|^2_{A^2(\Omega)} + C\sum_{n=1}^{N}\xi_n \\
\text{subject to} \quad & \xi_n \geq 0, \\
& t_n\Re[f(z_n)] \geq 1 - \xi_n,, \\
& \xi_n \geq \Im[f(z_n)]), \\
& \Im[f(z_n)] \geq -\xi_n
\end{aligned}
\tag{17}
$$

in which $\{\xi_n\}_{n=1}^N$ are slack variables introduced to allow for soft margins.

We now have the required tools and definitions to analyze the phenomenon from a geometrical perspective. In the next section, we will make use of the proposed framework to analyze the phenomenon in $A^2(\mathcal{X})$ and to find a robust classification learning rule for $A^2(\mathcal{X})$.

## 4 ROBUST CLASSIFICATION FOR HOLOMORPHIC HYPOTHESES

In this section, we will develop a robust learning rule with respect to $A^2(\mathcal{X})$. To do so, we first examine a toy problem to get an intuition for how adversarial examples occur in holomorphic hypotheses, and then continue to introduce the proposed robust learning rule.

To start, we will try to classify the unit disk $\mathbb{D}$ into distinct halves in which

$$
t(z) = \text{sign}(\Re[z]) \quad z \in \mathbb{D},
\tag{18}
$$

is the labeling function. We will choose our training samples to be the set

$$
S_n = \left\{ \left(z, t(z)\right) \mid z = e^{i\frac{k}{n}2\pi} \quad k = 0, \cdots, n-1 \right\}.
\tag{19}
$$

We will use the orthonormal polynomial basis of the unit disk as features,

$$
\varphi_k(z) = \sqrt{\frac{k+1}{\pi}} z^k.
\tag{20}
$$

We will visualize a holomorphic function in three ways. First, we use a domain coloring technique and graph the hypotheses on $\mathbb{D}$. The hue of a color represent the angle, and the saturation represent the magnitude of a complex number. We will also plot the contours of the real and imaginary parts of the hypotheses in the same graph using white and black lines respectively. Second, we will graph the real and the imaginary parts of the hypotheses on the unit circle $\mathbb{T} = \partial\mathbb{D}$. $\mathbb{T}$ is the set in which all the training points of $S_n$ are sampled from. In other words $\mathcal{S} = \mathbb{T}$. Third, we will graph the image of $\mathbb{T}$ in the range space of the hypotheses.

We have visualized the analogue of holomorphic optimal Bayes classifier $o_{\mathcal{D}}$ in the left column of figure 1. However, we have made use of the Szegő kernel to project the labeling function $t$ instead of the Bergman projection of $t$. The middle column of figure 1 depicts the output of applying the complex-valued SVC learning rule to $S_{30}$ using the orthonormal basis of $A^2(\mathbb{D})$ as features. As we have demonstrated in figure 1, the solution to program 17 is not robust with respect to $A^2(\mathbb{D})$. Nevertheless, due to the normality of the holomorphic functions, we can see that the learned hypothesis resembles $o_{\mathcal{D}}$, even though they are not learned through the same learning rule, or make use of the same hypothesis class for that matter.

Looking at figure 1, we can see that $o_{\mathcal{D}}$ is a transcendental function and has two logarithmic branch points on $i$ and $-i$. With this observation in mind, it is no wonder that approximating $o_{\mathcal{D}}$ is troublesome for our learning rule. Furthermore, we see that the essential singular region of the nonrobust

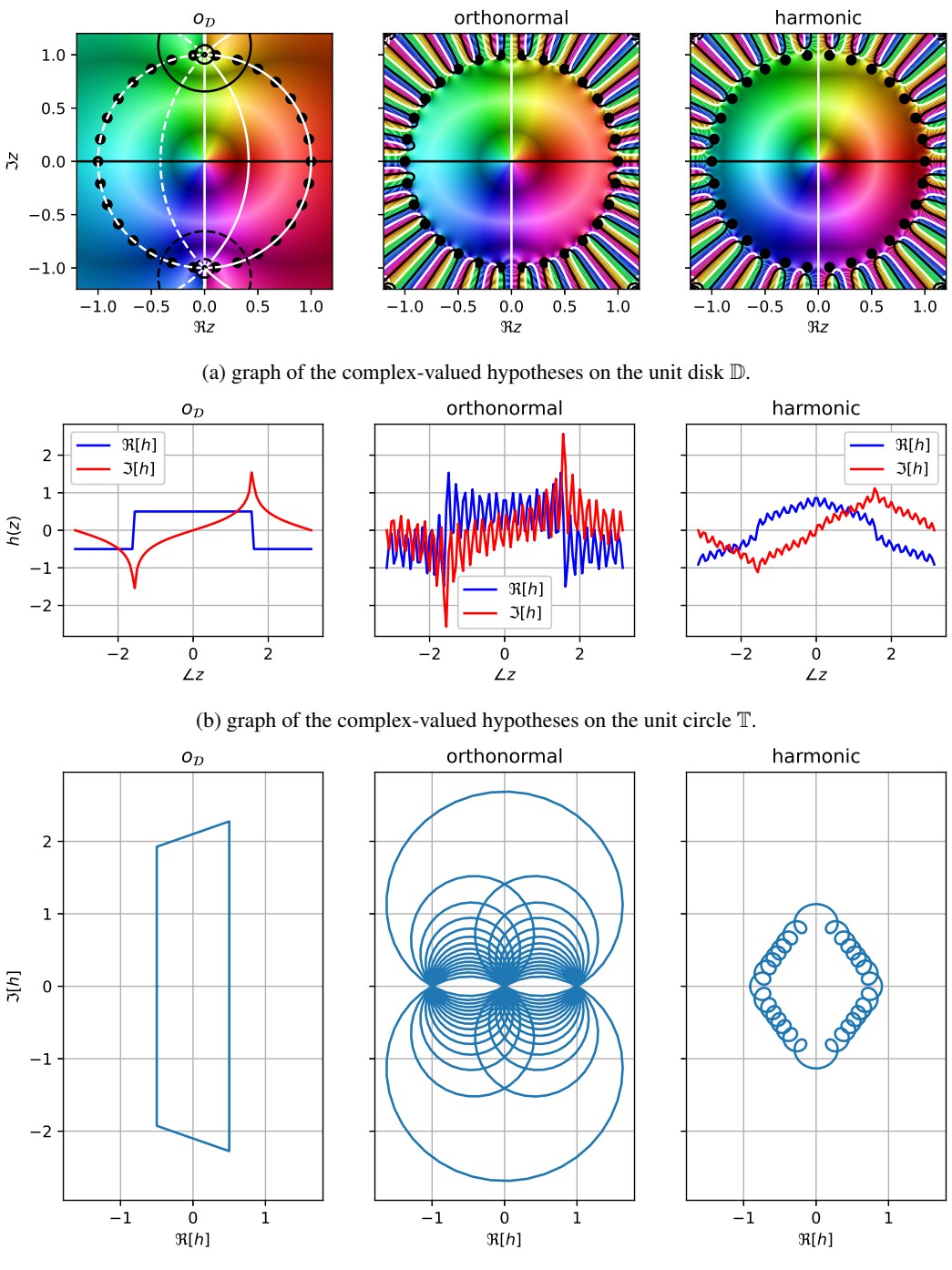

(a) graph of the complex-valued hypotheses on the unit disk $\mathbb{D}$.

(b) graph of the complex-valued hypotheses on the unit circle $\mathbb{T}$.

(c) image of $\mathcal{S} = \mathbb{T}$ under the complex-valued hypotheses.

Figure 1: Visualizations of $o_{\mathcal{D}}$ and the optimal hypotheses of the hinge loss function for $S_{30}$. We can see that even though the hypotheses are optimizing a different loss function than the 0-1 loss, they show certain similarities with $o_{\mathcal{D}}$.

hypothesis has advanced inside of $\mathbb{D}$. This advancement has caused the learners output to be nonrobust in a neighborhood of $\mathbb{T}$. This observation is common to nonrobust learning rules with respect to $A^2(\mathcal{X})$.

**Theorem 4.1.** *Let $h \in A^2(\mathcal{X})$ be the output of a nonrobust learning rule, then $h$ is robust on a dense open subset of $\mathcal{X}$.*

Theorem 4.1 shows that not only our choice of $A^2(\mathcal{X})$ as a replacement for $L^2(\mathcal{X})$ models the transferability of adversarial examples, it also correctly predicts the apparent paradox in the existence of the adversarial examples in ANNs, where the nonrobust hypothesis appears to be robust almost everywhere.

Figure 1c graphs the image of $\mathbb{T}$ in the range space of the hypotheses $h(\mathbb{T})$. Due to the singularities of the holomorphic projection of $t$, $o_\mathcal{D}(\mathbb{T})$ passes through the point at the infinity. Consequently, $o_\mathcal{D}(\mathbb{T})$ consists of two parallel lines; the trapezoidal shape in the figure is caused by the fact that we cannot in practice reach the point at the infinity. Looking at the nonrobust hypothesis in Figure 1c, we can see that the image of $\mathbb{T}$ passes through the decision boundary quite a few times. The figure shows that $h(\mathbb{T})$ of the nonrobust hypothesis is longer than necessary. In other words, it is possible to find a holomorphic hypothesis that achieves the same loss on the training set with $h(\mathbb{T})$ not passing through the decision boundary so many times. We can repeat the same argument for any other curve inside $\mathbb{D}$. Thus, we argue that if the output $h$ of $\mathcal{A}$ with respect to $A^2(\mathcal{X})$ minimizes the area covered by the image of $\mathcal{X}$ under $h$, then it is robust.

**Theorem 4.2.** *A learning rule $\mathcal{A}$ with respect to $A^2(\mathcal{X})$ that minimizes the Dirichlet energy of its output,*

$$E[h] = \int_\mathcal{X} \|\nabla h(z)\|^2 \, dV(z), \tag{21}$$

*is robust.*

Theorem 4.2 could be easily generalized to any differentiable hypothesis class. The reason is that the process of measuring the length of the image of a path is the same for all differentiable hypotheses; whether they are complex-valued or not.

**Definition 4.3** (robust (complex) SVC learning rule). The robust (complex) SVC learning rule is the same as the (complex) SVC learning rule, but minimizes $E[h]$ instead of $\|h\|^2_{\mathcal{H}(\mathcal{X})}$.

**Definition 4.4** (harmonic features). A set of non-constant features $\{\varphi_i\}_{i=1}^\infty$ are harmonic if

$$\int_\mathcal{X} \nabla \varphi_j(z) \cdot \nabla \varphi_k(z) \, dV(z) = \begin{cases} 1 & j = k \\ 0 & j \neq k \end{cases}. \tag{22}$$

Given the tuning matrix

$$\Sigma_{jk} = \int_\mathcal{X} \nabla \varphi_j(z) \cdot \nabla \varphi_k(z) \, dV(z), \tag{23}$$

we can transform any set of features to its harmonic counterpart,

$$\varphi^* = \Sigma^{-\frac{1}{2}} \varphi. \tag{24}$$

We emphasize that $\Sigma$ is positive definite by definition. Thus, a unique square root of $\Sigma$ exists and it is invertible.

**Theorem 4.5.** $\ell^2$ *regularization of a harmonic hypothesis minimizes its Dirichlet energy.*

We have repeated the toy experiment using the harmonic basis functions of $A^2(\mathcal{X})$, and reported the results in the right column of figure 1. We can see that the harmonic hypothesis is robust as expected. Thus, we have succeeded in finding a robust learning rule for $A^2(\mathcal{X})$. The interested reader can find more examples and experiments in the supplementary material of the paper.

## 5 ROBUST TRAINING OF ANNS

In this section, we will apply the results of section 4 to ANNs in a limited manner to demonstrate the feasibility and applicability of our approach. First, we put the proposed framework to use and describe how adversarial examples occur in ANNs.

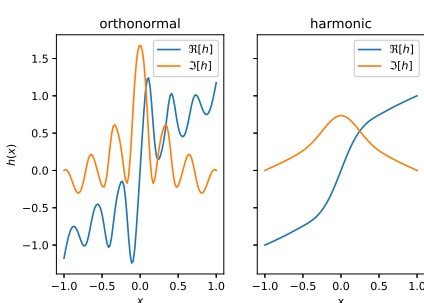

Figure 2: The output of the robust (right) and nonrobust (left) SVC learning rules for an integral representation with $\nu \in A^2(\mathbb{D})$ and $\sigma = H$.

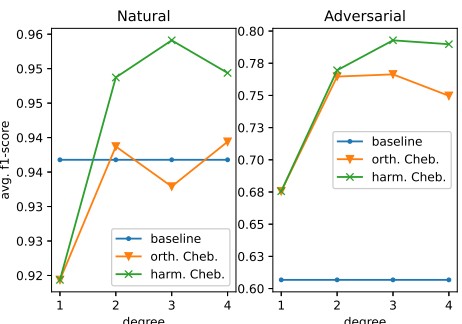

Figure 3: The result of performing a one-step $\ell^2$ normalized gradient-based black-box attack on harmonic and orthonormal Chebyshev polynomials for the UCI digits dataset. The baseline is a $64 \times 40 \times 30 \times 20 \times 10$ fully-connected MLP that is trained on the same dataset.

As stated in section 2, we represent neural networks using an integral representation. If we assume that $\nu \in A^2(\Omega)$, then we can repeat the process described in section 4 by replacing $\nu$ with its power series representation and follow a similar path. We already know that the orthonormal basis for $A^2(\Omega)$ would not be robust on the boundary of $\Omega$. In other words, the learning rule would fail to find the robust coefficients of neurons on the boundary of $\Omega$. When we use the harmonic bases of $A^2(\Omega)$ on the other hand, the region of convergence of $\nu(\omega)$ would cover $\partial\Omega$ and the output of the learning rule would be robust.

We have trained two neural networks which use the first 30 orthonormal and the harmonic bases to represent $\nu$. Both networks use the Heaviside activation function $H\big(\Re[\omega]x + \Im[\omega]\big) = \mathbf{1}_{\Re[\omega]x+\Im[\omega]>0}$. The interested reader can find the details of the experiment in the appendix D. We have reported the results of the experiment in figure 2. The figure suggests that the harmonic ANN is robust.

To showcase the applicability of the framework in a real-world scenario, we use a MLP classifier with ReLU activation as a surrogate and attack two polynomial hypotheses that are trained on a subset of the UCI digits dataset (Dua & Graff, 2017). We have reported the results in figure 3. The results show that the proposed robust learning rule is effective in mitigating the effects of the phenomenon, and supports the proposition that ANNs and polynomials are different representations of the same normal hypothesis class.

**Theorem 5.1.** *The output of the robust SVC learning rule with respect to a hypothesis $h$ represented in integral form satisfies the Poisson partial differential equation (PDE)*

$$-\Delta\nu(\omega) = \sum_{n=1}^{N} \lambda_n t_n \sigma(x_n; \omega), \tag{25}$$

*in which $\Delta$ is the Laplace operator.*

Theorem 5.1 shows that training a robust ANN is the same as solving a PDE. This observation suggests that robust classifiers are a normal family of functions. The fundamental solution $\Phi$ of the Laplace operator could be used to find $\nu$ as a function of $\{\lambda_n\}_{n=1}^{N}$,

$$\nu(\omega) = \sum_{n=1}^{N} \lambda_n t_n \int_{\Omega} \Phi(\omega - w)\sigma(x_n; w)\, dV(w). \tag{26}$$

**Definition 5.2** (harmonic activation function). A family of activation functions $\mathfrak{S}$ is harmonic if they satisfy the Helmholtz equation with the natural Dirichlet or Neumann boundary condition

$$-\Delta_\omega \mathfrak{s}(x; \omega) = \mathfrak{s}(x; \omega) \quad \mathfrak{s} \in \mathfrak{S}(\mathcal{X}). \tag{27}$$

**Theorem 5.3.** *$L^2$ regularization of $\nu(\omega)$ of a harmonic ANN is the same as minimizing its Dirichlet energy.*

Theorem 5.3 is the final result of this paper. We have managed to find a robust learning rule for a generalized notion of ANNs. Solving PDEs is an involved and complicated process, and we will not attempt to solve the derived PDEs in this paper. Nevertheless, both PDEs are very well-known and well-studied, and various methods and techniques are available in the literature which focuses on solving these PDEs.

## 6    CONCLUSION

In this paper we introduced a general framework for training robust ANN classifiers. We proposed a formal definition for robustness and transfer in learning theoretic terms, and described how adversarial examples might emerge from the pointwise convergence of the trained hypothesis to the infinite sample limit of the learning rule. Since our proposal assumes that the convergence is pointwise, we need a separate explanation for the transfer of adversarial examples between hypotheses under pointwise convergence. To this end, we propose the normal hypothesis classes, and define these classes by adapting the definition of a normal family of functions. We also introduce integral representations as an abstraction of ANNs that is easier to analyze in terms of convergence.

Next, we showed that under the proposed definitions, learning rules for ANNs that are converging to a hypothesis in $L^2(\mathcal{X})$ does not explain the observation of transfer between different architectures of ANNs. As an alternative, we propose that transfer in ANNs is better modeled by functions in $\mathcal{O}(\mathcal{X})$. $\mathcal{O}(\mathcal{X})$ is the set of holomorphic functions on $\mathcal{X}$. We provide the necessary definitions to use $\mathcal{O}(\mathcal{X})$ as a hypothesis class, and take the first steps to enable the use of powerful tools of complex analysis in the study of the phenomenon.

Through holomorphicity, our framework provides a geometrical interpretation for the adversarial examples phenomenon. We conclude that a binary classifier with minimal Dirichlet energy is robust. In other words, replacing the $L^2$ regularization term in the loss function with the Dirichlet energy of the hypothesis should result in a robust classifier. Minimizing the Dirichlet energy might not be tractable in gradient descent as we need to compute the derivative of the Dirichlet energy to do so. To circumvent this problem, we introduce the harmonic features and activation functions. In summary, we first construct features or activation functions that satisfy a condition, and then show that $L^2$ regularization of these hypothesis classes is the same as minimizing their Dirichlet energy. Consequently, minimizing the Dirichlet energy for these hypotheses could be as efficient as $\ell^2$ regularization of their parameters.

To the best of our knowledge, we have provided the first method that makes use of calculus of variations and differential equations to tackle the challenge of robust training of ANNs. Nevertheless, the analysis proved that training robust ANNs classifiers is not trivial, and we need to either implicitly or explicitly solve an intermediate PDE. There are multiple methods of doing so available in the relevant literature. As a result, we have left a large-scale implementation of the proposed method to future, when an in depth understanding of the appropriate parameter domain sets and harmonic activation functions in ANNs is achieved.

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

## A   ROBUST TRAINING IN A NUTSHELL

The main text is concerned with the theoretical aspects of the proposed framework. While the theoretical discussion is necessary, some readers might only be interested in the implementation of the framework. Here, we will describe the framework in a step-by-step manner from a practical perspective.

The goal in our framework is to show that a learning rule $\mathcal{A}$ satisfies the condition in definition 2.2. In other words, we want to guarantee that $h_n = \mathcal{A}(d_n)$ converges uniformly to $h = \mathcal{A}(\mathcal{D})$. For example, Barati et al. (2021) find a robust learning rule for polynomials by choosing a particular sequence of training points (the Chebyshev grid) that guarantee the uniform convergence of the sequence of hypotheses. However, finding optimal training sets in practice is not a well-defined process. Ilyas et al. (2019) need a robust classifier in the first place to generate an optimal training

set and applying the proposal of Barati et al. (2021) to ANNs does not end up with a unique set of optimal training points as it did with polynomials.

In our framework, theorem 4.2 provides an alternative to an optimal sequence of training points. In this case, $h_n : \mathcal{X} \to \mathbb{C}$ would map $\mathcal{X}$ to a subset of $\mathbb{C}$ with minimal area. The area of $h_n(\mathcal{X})$ is decided by $d_n$ and it will be less than or equal to the area of $h(\mathcal{X})$. We exploit this fact to guarantee uniform convergence of $\{h_n\}_{n=1}^{\infty}$ for binary classifiers. It is conceivable that a similar approach could be taken for other applications of machine learning as well.

In conclusion, implement the proposed framework in the following steps:

1. Check if $\mathcal{A}$ satisfies definition 2.2. If it does, $\mathcal{A}$ is already robust. For example, Barati et al. (2021) used Bernstein polynomials to uniformly approximate the optimal hypothesis.
2. Replace the regularization term of $\mathcal{A}$ with Dirichlet energy of the hypothesis $E[h_n]$.
3. Check if the updated $\mathcal{A}$ satisfies definition 2.2. To do so, consider an arbitrary path $\gamma : \mathbb{R} \to \mathcal{X}$ and prove that $h$ is robust on $\gamma$ when length of $h(\gamma)$ is minimal.
4. If computing $E[h_n]$ is intractable, consider finding a set of suitable harmonic features.

## B  COMPARISON WITH THE LITERATURE

The main point of distinction between the proposed framework and the competing proposals is in the use of complex analysis to simplify the analysis. Even though analytic properties of the hypotheses has been exploited before in the literature to derive various methods of defence and attack, e.g. Wen (2022), the use of holomorphic functions and complex analysis in machine learning have been sparse Barkatou & Jaroschek (2018); Heilman et al. (2015); Sarma et al. (2019). We believe the reason to be that applying complex analysis to machine learning is an interdisciplinary effort by nature.

Goodfellow et al. (2015) were the first to show that nonrobust ANNs are weak to analytic attacks, and attributed the phenomenon to something that they called linearity. Tanay & Griffin (2016) refuted this claim by showing that linear classifiers could be robust. We answer the apparent paradox between linear and nonlinear nature of the phenomenon by proposing that both positions are in some sense correct, and that the phenomenon would be better described by the analytic properties of the robust hypothesis.

Hein & Andriushchenko (2017) proposed a certificate of robustness for differentiable classifiers. The same certificate could be used for analytic functions. The certificate might be improved by considering the Taylor series expansion of the classifier around a test sample. The Abel-Ruffini theorem might result in some complications in this process.

From a geometric point of view, our proposal is very well aligned with the dimpled manifold model of Shamir et al. (2021). Regarding the boundary tilting perspective of Tanay & Griffin (2016), it seems that the nonrobust classifier in figure 1c has managed to cross the decision boundary at right angles, and there seems to be exceptions to the boundary tilting perspective in nonlinear cases. Paknezhad et al. (2021) hypothesises that the robust classifier would maximize the margin. We did not make an attempt to analyze the learning rule from the perspective of maximum margin classification. Nonetheless, the peculiar shape of the image of $\mathbb{T}$ under the robust hypothesis in figure 1c suggests that the proposed robust SVC learning rule also maximizes the margin in some sense.

Our model for the transfer of adversarial examples is mostly similar to the proposal of Goodfellow et al. (2015), in which the reason for the transfer of adversarial examples are deemed to be that the models converge to the optimal linear hypothesis. Papernot et al. (2016) shows that existing machine learning approaches are in general vulnerable to systematic black-box attacks regardless of their structure, showing that transferable adversarial examples are common in machine learning models. Here, we have defined the holomorphic optimal Bayes classifier and given a formal definition of how transfer occurs. Ilyas et al. (2019) moves the blame to hidden patterns in the input, which our proposal does not align with. Inkawhich et al. (2019) relates the transfer of the adversarial examples to the learned features as well.

There are multiple instances of the use of gradient information through out the literature (Ros & Doshi-Velez, 2018; Paknezhad et al., 2021). The main contrast between our proposal is that we

manage to relate gradient regularization with the geometry of $f(\mathcal{X})$ and further reveal how gradient regularization is related with the Dirichlet energy of the hypothesis.

## C  PROOF OF THE THEOREMS

### C.1  THEOREM 2.3

*Proof.*  $L^2(\mathcal{X})$ would be nonuniform learnable if and only if it is a countable union of PAC learnable hypothesis classes $\mathcal{H}_\alpha(\mathcal{X})$. Since $L^2(\mathcal{X})$ is a Hilbert space, it has an orthonormal basis $\{\varphi_\alpha\}_{\alpha=0}^\infty$. Then, we can choose

$$\mathcal{H}_\alpha(\mathcal{X}) = \{ \sum_{k=0}^{\alpha} a_k \varphi_k(x) | a_k \in \mathbb{R} \}, \tag{28}$$

and the theorem would follow. $\square$

### C.2  THEOREM 2.5

*Proof.*  The Lagrangian of program 7 is

$$\mathcal{L} = \frac{1}{2} \int_\Omega |\nu(\omega)|^2 \, dV(\omega) + \sum_{n=1}^{N} \lambda_n \big( 1 - t_n \int_\Omega \nu(\omega)\sigma(x_n;\omega) \, dV(\omega) \big). \tag{29}$$

We can rearrange the Lagrangian into

$$\mathcal{L} = \int_\Omega \frac{1}{2} |\nu(\omega)|^2 - \sum_{n=1}^{N} \lambda_n t_n \nu(\omega)\sigma(x_n;\omega) \, dV(\omega) + \sum_{n=1}^{N} \lambda_n, \tag{30}$$

$$= \int_\Omega L(\omega, \nu, \nabla\nu) \, dV(\omega) + \sum_{n=1}^{N} \lambda_n. \tag{31}$$

By the Euler-Lagrange equations for a function of several variables, the optimal $\nu$ must satisfy the following PDE

$$\frac{\delta L}{\delta \nu} - \sum_{j=1}^{d} \frac{\partial}{\partial x_j} \frac{\delta L}{\delta \nu_j} = 0, \tag{32}$$

in which $\nu_j = \frac{\partial \nu}{\partial x_j}$, and $\frac{\delta}{\delta \nu}$ is the usual differentiation with the exception that it treats $\nu$ like a symbolic variable. Thus,

$$\nu(\omega) = \sum_{n=1}^{N} \lambda_n t_n \sigma(x_n;\omega). \tag{33}$$

Similarly, the Lagrangian of program 6 is

$$\mathcal{L} = \frac{1}{2} \sum_{i=0}^{\infty} a_i^2 + \sum_{n=1}^{N} \lambda_n \big( 1 - t_n \sum_{i=0}^{\infty} a_i \varphi_i(x_n) \big). \tag{34}$$

The optimal $a_i$ is achieved when the derivative of the Lagrangian with respect to $a_i$ vanishes. Thus,

$$a_i = \sum_{n=1}^{N} \lambda_n t_n \varphi_i(x_n). \tag{35}$$

To see that the learning rule is not robust in general, consider the Dirac's delta, which could be constructed by taking the limit of a Normal probability distribution function with vanishing variance,

$$\delta(x;\omega) = \lim_{\sigma \to 0} \mathcal{N}(x;\omega,\sigma) = \begin{cases} \infty & x = \omega \\ 0 & \text{otherwise} \end{cases}. \tag{36}$$

We can see that the output of the SVC learning rule,

$$h(x) = \int_\Omega \nu(\omega)\delta(x;\omega)\,dV(\omega) = \nu(x), \tag{37}$$

$$= \sum_{n=1}^{N} \lambda_n t_n \delta(x_n; x), \tag{38}$$

would be the same as the memoising learning rule, which is the text book example of a consistent learner that is not nonuniform. Thus, the sample complexity of the SVC learning rule is dependent on the distribution of training samples. Our definition of robustness is the same as Cauchy's criterion for uniform convergence of a sequence of functions. Since the sample complexity of the SVC learning rule depends on the distribution of training samples, $\{\mathcal{A}(d_n)\}_{n=1}^\infty$ cannot be uniformly converging to its limit in general.

$\square$

### C.3 PROPOSITION 2.6

*Proof.* Consider the integral representation

$$h(x) = \int_\Omega \nu(\omega)\delta(x;\omega)\,dV(\omega) = \nu(x). \tag{39}$$

According to theorem 2.5, in the infinite sample limit we would have

$$h(x) = \int_{\mathcal{X}} \lambda(\zeta)t(\zeta)\delta(x;\zeta)\,dV(\zeta), \tag{40}$$

$$= \lambda(x)t(x). \tag{41}$$

Since the norm of $h$ should be minimal and $h$ should satisfy $t(x)h(x) \geq 1$, we can deduce that $\lambda(x) = 1$ for all $x$. We can ignore the samples on the decision boundary since they are a degenerate subset of $\mathcal{X}$. Consequently, if $t \in \mathcal{H}$, then $t$ would remain feasible and would dominate all other $h \in \mathcal{H}$ in terms of the training loss and the regularization score. $\square$

### C.4 THEOREM 2.9

*Proof.* Since $\mathcal{H}$ is normal, we know that for large and diverse enough training sets it is true that

$$\|h - \hat{h}\|_\infty \leq \epsilon. \tag{42}$$

Moreover, since $\partial\mathcal{H}$ is normal as well, it is true that

$$\|\frac{\partial\hat{h}}{\partial x_i} - \frac{\partial h}{\partial x_i}\|_\infty \leq \epsilon'. \tag{43}$$

Given both conditions, we can infer that a gradient based attack would follow a similar path for both $h$ and $\hat{h}$ and the label that would be assigned to the generated adversarial example would also be probably approximately equal. $\square$

### C.5 THEOREM 2.10

*Proof.* It is enough to find a learning rule $\mathcal{A}$ with respect to $L^2(\mathcal{X})$ that does not satisfy the condition for being normal. The SVC learning rule for the integral representation

$$h(x) = \int_\Omega \nu(\omega)\delta(x;\omega)\,dV(\omega) \tag{44}$$

is such a learning rule. To see why, imagine a set of training points $S \sim d_n$. Then $h = \mathcal{A}(S)$ would be a point mass function. Consequently, we cannot find a dense set $K \subseteq \mathcal{X}$ for which

$$\|\mathcal{A}(d_n) - \mathcal{A}(\mathcal{D})\|_\infty \leq \epsilon. \tag{45}$$

Thus, $L^2(\mathcal{X})$ is not a normal hypothesis class. $\square$

## C.6 Theorem 3.5

*Proof.* First, assume that $\mathcal{A}$ is a nonuniform learner with respect to $A^2(\mathcal{X})$. By the definition of nonuniform learnability, for any $\epsilon, \delta > 0$ a natural number $N$ exists that for any training set $S$ larger than $N$ and with probability $1 - \delta$ we have that

$$\text{loss}(t(x), x, \mathcal{A}(S)) \leq \epsilon. \tag{46}$$

Then every $\{\mathcal{A}(d_n)\}_{n=1}^{\infty}$ is converging uniformly to $\mathcal{A}(\mathcal{D})$.

**Proposition C.1.** *Suppose $\{f_n\}_{n=1}^{\infty} \subset \mathcal{O}(\mathcal{X})$ converges uniformly on compact subsets of $\mathcal{X}$ to the function $f : \mathcal{X} \to \mathbb{C}$. Then $f \in \mathcal{O}(\mathcal{X})$ and for each $\alpha \in \mathbb{N}^d$,*

$$\lim_{n\to\infty} \partial^\alpha f_n = \partial^\alpha f \tag{47}$$

*compactly in $\mathcal{X}$. $\partial^\alpha$ is a shorthand notation for*

$$\frac{\partial^{|\alpha|}}{\partial x_1^{\alpha_1} \partial x_2^{\alpha_2} \cdots \partial x_d^{\alpha_d}}, \quad |\alpha| = \sum_{j=1}^{d} \alpha_j. \tag{48}$$

Thus, we can deduce that

$$\|\frac{\partial \mathcal{A}(d_n)}{\partial x_j} - \frac{\partial \mathcal{A}(\mathcal{D})}{\partial x_j}\|_\infty \leq \epsilon \quad j = 1, \cdots, d. \tag{49}$$

Equation 49 shows that $A^2(\mathcal{X})$ and $\partial A^2(\mathcal{X})$ are normal hypothesis classes with respect to nonuniform learners.

**Proposition C.2** (Krantz (2010)). *Let $\{f_n\}_{n=1}^{\infty}$ be a sequence of holomorphic functions on a domain $\mathcal{X} \subset \mathbb{C}^d$. Assume that the sequence converges pointwise to a limit function $f$ on $\mathcal{X}$. Then $f$ is holomorphic on a dense open subset of $\mathcal{X}$. Also the convergence is uniform on compact subsets of the dense open set.*

Proposition C.2 shows that when $\mathcal{A}$ is a consistent learner for $A^2(\mathcal{X})$ and $\{\mathcal{A}(d_n)\}_{n=1}^{\infty}$ is converging pointwise to $\mathcal{A}(\mathcal{D})$, then the convergence is uniform on some dense open $K \subset \mathcal{X}$ and we have that

$$\|\mathcal{A}(d_n) - \mathcal{A}(\mathcal{D})\|_\infty \leq \epsilon \tag{50}$$

$$\|\frac{\partial \mathcal{A}(d_n)}{\partial x_j} - \frac{\partial \mathcal{A}(\mathcal{D})}{\partial x_j}\|_\infty \leq \epsilon' \tag{51}$$

on $K$. As a result, $A^2(\mathcal{X})$ and $\partial A^2(\mathcal{X})$ are normal hypothesis classes with respect to universally consistent learners. $\qquad\square$

## C.7 Theorem 4.1

*Proof.* Since $\mathcal{A}$ is not robust, $\{\mathcal{A}(d_n)\}_{n=1}^{\infty}$ is a pointwise converging sequence of holomorphic functions. According to proposition C.2, $\mathcal{A}(\mathcal{D})$ is holomorphic on a dense open subset $K \subset \mathcal{X}$ and

$$\|\mathcal{A}(\mathcal{D}) - o_{\mathcal{D}}\|_\infty \leq \epsilon \tag{52}$$

on $K$. $\qquad\square$

## C.8 Theorem 4.2

*Proof.* Consider a smooth path $\gamma : [0, 1] \to \mathcal{X}$. The length of the image of $\gamma$ under $h \in A^2(\mathcal{X})$ is

$$\|h(\gamma)\| = \int_0^1 \|\nabla h(\gamma(t))\| \gamma'(t) \, dt, \tag{53}$$

Hence, when $\mathcal{A}$ has minimized $E[h]$, it has minimized $\|h(\gamma)\|$ for all $\gamma$. Consequently, the image of all of the paths that start from a training sample $z$ would stay as close to $f(z)$ as possible. It follows

that $h(\mathcal{S})$ and the imaginary axis, which represents the decision boundary, would have the minimum number of intersection points allowed by the training set $S$.

According to definition 2.2, to show that $\mathcal{A}$ is robust, we have to show that for any $\epsilon \geq 0$ a non-negative integer $N$ exists for which the output of $\mathcal{A}$ for any two training set $S, S' \sim \mathcal{D}$ larger than $N$ would satisfy

$$\|\mathcal{A}(S) - \mathcal{A}(S')\|_\infty \leq \epsilon. \tag{54}$$

Without loss of generality, assume that $\mathcal{A}(\mathcal{D})$ partitions $\mathcal{X}$ into $M$ regions. Thus, as long as $S$ is diverse enough so that we have at least one sample from each region, the number of times that the image of any curve $\mathcal{A}(\mathrm{d}_n)(\gamma)$ intersect with the decision boundary would not change. Consequently, adding any more samples to the training set would only results into a more accurate estimation of the position of the decision boundary. In other words, it is always the case that after adding the required $M$ support vectors to $S_n$, the position of the decision boundary would not make any drastic changes. As a result, for big enough training sets, equation 54 would be satisfied.

Keep in mind that the same could not be said about a learning rule that does not minimize $E[h]$ since there is no guarantee that $\mathcal{A}(\mathrm{d}_n)(\gamma)$ and the decision boundary does not intersect more than necessary. □

## C.9 Theorem 4.5

*Proof.* $h$ has a representation

$$h(z) = \sum_{\alpha \geq 0} \overline{a_\alpha} \varphi_\alpha(z) = a^H \varphi(z). \tag{55}$$

We can prove the theorem by simply calculating $E[f]$,

$$\int_\Omega \|\nabla h(z)\|^2 \, dV(z) = \int_\Omega a^H J(z) J(z)^H a \, dV(z), \tag{56}$$

$$= a^H \int_\Omega J(z) J(z)^H \, dV(z) a, \tag{57}$$

$$= a^H \Sigma a = a^H a, \tag{58}$$

in which $J(z)$ is the Jacobian of the feature vector $\varphi(z)$. □

## C.10 Theorem 5.1

*Proof.* The Lagrangian for the real-valued robust SVC learning rule is

$$\mathcal{L} = \frac{1}{2} \int_\Omega \|\nabla\nu(\omega)\|^2 \, dV(\omega) + \sum_{n=1}^N \lambda_n \big(1 - t_n \int_\Omega \nu(\omega)\sigma(x_n;\omega) \, dV(\omega)\big), \tag{59}$$

$$= \int_\Omega \frac{1}{2} \sum_{j=1}^d \nu_j(\omega)^2 - \sum_{n=1}^N \lambda_n t_n \nu(\omega)\sigma(x_n;\omega) \, dV(\omega) + \sum_{n=1}^N \lambda_n, \tag{60}$$

$$= \int_\Omega L(\omega, \nu, \nabla\nu) \, dV(\omega) + \sum_{n=1}^N \lambda_n. \tag{61}$$

By the Euler-Lagrange equations for a function of several variables, $\nu$ satisfies the following equation

$$\frac{\delta L}{\delta \nu} - \sum_{j=1}^d \frac{\partial}{\partial \omega_j} \frac{\delta L}{\delta \nu_j} = 0. \tag{62}$$

Consequently, $\nu$ must satisfy the following PDE

$$-\Delta\nu(\omega) = \sum_{n=1}^N \lambda_n t_n \sigma(x_n;\omega). \tag{63}$$

□

## C.11 THEOREM 5.3

*Proof.* We know that

$$\nu(\omega) = \sum_{n=1}^{N} a_n \mathfrak{s}_n(\omega), \tag{64}$$

$$-\Delta \mathfrak{s}_n(\omega) = \mathfrak{s}_n(\omega). \tag{65}$$

Computing $E[\nu]$, we would have

$$E[\nu] = \int_{\Omega} \nabla \nu(\omega) \cdot \nabla \nu(\omega) \, dV(\omega), \tag{66}$$

$$= \sum_{n=1}^{N} \sum_{m=1}^{N} a_n a_m \int_{\Omega} \nabla \mathfrak{s}_n(\omega) \cdot \nabla \mathfrak{s}_m(\omega) \, dV(\omega). \tag{67}$$

Since $\mathfrak{s}$ is twice differentiable, we may apply the divergence theorem to get

$$\int_{\Omega} \nabla \cdot \big(\mathfrak{s}_n(\omega) \nabla \mathfrak{s}_m(\omega)\big) \, dV(\omega) = \int_{\Omega} \nabla \mathfrak{s}_n(\omega) \cdot \nabla \mathfrak{s}_m(\omega) - \mathfrak{s}_n(\omega)\mathfrak{s}_m(\omega) \, dV(\omega), \tag{68}$$

$$= \int_{\partial \Omega} \mathfrak{s}_n(\omega)(\nabla \mathfrak{s}_m \cdot \hat{n}) \, dS(\omega), \tag{69}$$

where $\hat{n}$ is the outward-pointing normal of the boundary of $\Omega$. When either $\mathfrak{s}_m$ satisfies the natural Newman or $\mathfrak{s}_n$ satisfy the natural Dirichlet boundary conditions, the right hand side of equation 68 must vanish. Consequently,

$$\int_{\Omega} \nabla \mathfrak{s}_n(\omega) \cdot \nabla \mathfrak{s}_m(\omega) \, dV(\omega) = \int_{\Omega} \mathfrak{s}_n(\omega)\mathfrak{s}_m(\omega) \, dV(\omega). \tag{70}$$

By replacing the expression in $E[\nu]$ we would have

$$E[\nu] = \|\nu\|_{L^2(\mathcal{X})}^2 = \sum_{n=1}^{N} \sum_{m=1}^{N} a_n a_m \int_{\Omega} \mathfrak{s}_n(\omega)\mathfrak{s}_m(\omega) \, dV(\omega). \tag{71}$$

$\square$

# D DETAILS OF THE EXPERIMENTS

## D.1 SECTION 4

First, we will find $o_{\mathcal{D}}$ by projecting $t(z)$ into $H^2(\mathbb{T}) \subset L^2(\mathbb{T})$,

$$o_{\mathcal{D}}(z) = \int_{\mathbb{T}} t(\zeta) S_{\mathbb{T}}(z, \zeta) \, d\zeta. \tag{72}$$

$H^2(\mathbb{T})$ is the Hardy space on $\mathbb{T}$ and its reproducing kernel is the Szegő kernel $S_{\mathbb{T}}(z, \zeta)$. The theory of the Bergman and the Szegő kernels are identical as far as we are concerned. Consequently,

$$o_{\mathcal{D}}(z) = \int_{-\pi}^{\pi} t(e^{i\theta}) \frac{1}{2\pi(1 - ze^{-i\theta})} \, d\theta, \tag{73}$$

$$= \frac{1}{2\pi} \int_{-\pi}^{\pi} \frac{\operatorname{sign}(\cos\theta)}{1 - ze^{-i\theta}} \, d\theta, \tag{74}$$

$$= \frac{1}{2\pi} \Big( \int_{-\frac{\pi}{2}}^{\frac{\pi}{2}} \frac{1}{1 - ze^{-i\theta}} \, d\theta - \int_{\frac{\pi}{2}}^{\frac{3\pi}{2}} \frac{1}{1 - ze^{-i\theta}} \, d\theta \Big), \tag{75}$$

$$= \frac{i}{\pi} \Big( \log(-z - i) - \log(-z + i) \Big). \tag{76}$$

The projection has two logarithmic branch points and our computer algebra system chooses the branch cuts to be parallel to the real axis. We rotate them by 90 degrees in opposite directions so that the branch cuts are perpendicular to $\mathbb{T}$ and fall outside of $\mathbb{D}$. To rotate the branch cuts by an angle $\phi$, we have to multiply the expression inside the logarithms by $e^{i\phi}$. Thus, the projection of $t(z)$ into $H^2(\mathbb{T})$ is

$$o_{\mathcal{D}}(z) = \frac{i}{\pi}\big(\log(i(-z-i)) - \log(-i(-z+i))\big), \tag{77}$$

$$= \frac{i}{\pi}\big(\log(1-iz) - \log(1+iz)\big). \tag{78}$$

Next, we will find a non-robust hypothesis in $A^2(\mathbb{D})$. As we have described in the paper, the orthonormal bases for $A^2(\mathbb{D})$ are

$$\varphi_k(z) = \frac{z^k}{\sqrt{\gamma_k}}, \tag{79}$$

$$\gamma_k = \int_{\mathbb{D}} |z|^{2k}\, dV(z) = \frac{\pi}{k+1}. \tag{80}$$

We choose a series representation for our hypothesis,

$$h(z) = b + \sum_{k=1}^{K} a_k \varphi_k(z), \tag{81}$$

then use a convex optimization library to solve program 17 for $h$ with $S_{30}$ as the training set.

Finally, we train a robust hypothesis with the help of theorem 4.5. To do so, we need to find the harmonic bases for $A^2(\mathbb{D})$. The elements of the tuning matrix $\Sigma_{jk}$ of the polynomial bases are

$$\Sigma_{jk} = \int_{\mathbb{D}} \frac{d}{dz}z^j \overline{\frac{d}{dz}z^k}\, dV(z), \tag{82}$$

$$= jk \int_{\mathbb{D}} z^{(j-1)}\overline{z}^{(k-1)}\, dV(z), \tag{83}$$

$$= \begin{cases} 0 & j \neq k \\ k\pi & j = k \end{cases}. \tag{84}$$

Consequently, the harmonic bases of $A^2(\mathbb{D})$ are

$$\varphi_k^*(z) = \frac{z^k}{\sqrt{k\pi}}. \tag{85}$$

The reader might find it interesting that the corresponding kernel for harmonic bases of $A^2(\mathbb{D})$ is the polylogarithm of order 1,

$$K_{\mathbb{D}}^*(z,\zeta) = \sum_{k=1}^{\infty} \varphi_k^*(z)\overline{\varphi_k^*(\zeta)}, \tag{86}$$

$$= \frac{1}{\pi}\sum_{k=1}^{\infty} \frac{(z\overline{\zeta})^k}{k}, \tag{87}$$

$$= -\frac{1}{\pi}\log(1 - z\overline{\zeta}). \tag{88}$$

## D.2 SECTION 5

We train a single-layer perceptron in the first part of section 5. The training samples are a set of 8 equispaced points in $[-1, 1]$ and are labeled by their sign. We choose an integral representation for the network,

$$h(x) = \int_{\mathbb{D}} \nu(\omega)H\big(\Re[\omega]x + \Im[\omega]\big)\, dV(\omega), \quad v \in A^2(\mathbb{D}), \tag{89}$$

In which $H(x) = 1_{x>0}$ is the Heaviside step function. We need to choose a representation for $\nu$ to compute $h$. In this experiment we want to compare the performance of the orthonormal and harmonic bases of $A^2(\mathbb{D})$. Replacing $\nu$ with the corresponding representations we have

$$h(x) = \sum_{k=0}^{\infty} a_n \int_{\mathbb{D}} \frac{\omega^k}{\sqrt{\gamma_k}} H\big(\Re[\omega]x + \Im[\omega]\big) \, dV(\omega), \tag{90}$$

$$= \sum_{k=0}^{\infty} a_k \frac{\psi_k(x)}{\sqrt{\gamma_k}}. \tag{91}$$

Thus, the harmonic and orthonormal $h$ have representations

$$h(x) = \sum_{k=0}^{\infty} a_k \sqrt{\frac{k+1}{\pi}} \psi_k(x), \tag{92}$$

$$h^*(x) = \sum_{k=0}^{\infty} a_k \sqrt{\frac{1}{k\pi}} \psi_k(x). \tag{93}$$

Next, we will compute $\psi_k$

$$\psi_k(x) = \int_{\mathbb{D}} \omega^k H\big(\Re[\omega(x-i)]\big) \, dV(\omega), \tag{94}$$

$$= \int_{\Re[\omega(x-i)]>0} \omega^k \, dV(\omega), \tag{95}$$

$$= \int_{-\tan^{-1}(x)}^{-\tan^{-1}(x)+\pi} \int_0^1 r^k e^{ik\theta} \, r dr d\theta, \tag{96}$$

$$= \frac{1}{(k+2)} \int_{-\tan^{-1}(x)}^{-\tan^{-1}(x)+\pi} e^{ik\theta} \, d\theta, \tag{97}$$

$$= \frac{-i e^{ik(-\tan^{-1}(x))}}{k(k+2)} \big((-1)^k - 1\big), \tag{98}$$

$$= \begin{cases} 0 & k \text{ even} \\ \frac{-i \exp\big(-ik\tan^{-1}(x)\big)}{k(k+2)} & k \text{ odd} \end{cases}. \tag{99}$$

Finally, we will train $h$ and $h^*$ using the complex SVC learning rule. If the norm of $\psi$ was too small and training was numerically unstable, multiplying all $\psi_k$ with a constant would not change the results.

The second experiment compares the robustness of a MLP with the robustness of harmonic and orthonormal Chebyshev bases. The MLP size is $64 \times 40 \times 30 \times 20 \times 10$ and all the neurons are ReLU activated. The MLP has minimized the cross entropy loss and is trained by the ADAM optimizer.

We will use the Chebyshev bases $T_\alpha$ as polynomial features. However, it would be computationally intractable to to include all possible polynomial bases in the hypothesis. To see why, consider the count of all polynomial bases where each $x_i$ has a degree of at most one. This set has the same cardinality with a set that contains all possible subsets of $\{x_i\}_{i=1}^{64}$, i.e. the power set of $\{x_i\}_{i=1}^{64}$. Consequently, we have to decide which polynomial basis would be added to the hypothesis.

To do so, we enumerate all the possible walks up to $D$ steps on a 8x8 lattice, and then map each walk to a polynomial. A node in the lattice represents a pixel, and the structure of the lattice represents the structure of a 2D image. A walk with $D$ steps would be mapped to a degree $D$ polynomial. For example, the following walk

$$x_{00} \to x_{01} \to x_{11} \to x_{10} \to x_{00} \tag{100}$$

would be mapped to

$$T_2(x_{00})T_1(x_{01})T_1(x_{11})T_1(x_{10}). \tag{101}$$

We add a loop to every node in the lattice so that monomials would be included in the set of features as well.

Next, we have to compute the elements of the tuning matrix $\Sigma_{\alpha\beta}$,

$$\Sigma_{\alpha\beta} = \int_{[-1,1]^{64}} \nabla(T_{\alpha_1}(x_1)\cdots T_{\alpha_{64}}(x_{64})) \cdot \nabla(T_{\beta_1}(x_1)\cdots T_{\beta_{64}}(x_{64})) \, dV(x), \tag{102}$$

$$= \int_{[-1,1]^{64}} \sum_{j=1}^{64} \frac{d}{dx_j} T_{\alpha_j}(x_j) \frac{d}{dx_j} T_{\beta_j}(x_j) \prod_{j\neq k} T_{\alpha_k}(x_k) T_{\beta_k}(x_k) \, dV(x), \tag{103}$$

$$= \int_{[-1,1]^{64}} \sum_{j=1}^{64} \alpha_j \beta_j U_{\alpha_j-1}(x_j) U_{\beta_j-1}(x_j) \prod_{j\neq k} T_{\alpha_k}(x_k) T_{\beta_k}(x_k) \, dV(x). \tag{104}$$

We need the following formulas for computing the integrals,

$$U_n(x) = \begin{cases} 2\sum_{\text{odd } j}^n T_j(x) & \text{for odd } n. \\ 2\sum_{\text{even } j}^n T_j(x) - 1 & \text{for even } n, \end{cases} \tag{105}$$

$$\int_{-1}^1 T_n(x)\, T_m(x) \, \frac{dx}{\sqrt{1-x^2}} = \begin{cases} 0 & \text{if } n \neq m, \\ \pi & \text{if } n = m = 0, \\ \frac{\pi}{2} & \text{if } n = m \neq 0. \end{cases} \tag{106}$$

Finally, we compute the harmonic Chebyshev bases and train the polynomial hypotheses using the SVC learning rule. To test the robustness of the trained polynomials, we evaluate the performance of the polynomials on adversarial examples that are generated by attacking the MLP with a single-step $\ell_2$ normalized gradient-based attack.

## E  DOMAINS OF HOLOMORPHY

In this section, we provide a brief summary of domains of holomorphy and describe how such domains could be constructed for common applications of machine learning. In complex analysis, the main subject of study are holomorphic functions of a scalar $z \in \mathbb{C}$. Domains of holomorphic functions in this setting are simple objects that are described by the Riemann mapping theorem and its generalizations, which states that if $\mathcal{X}$ is a simply connected domain in $\mathbb{C}$ and is not $\mathbb{C}$ itself, then a biholomorphic map between $\mathcal{X}$ and the unit disk $\mathbb{D}$ exists. One way to show that some domain is a domain of holomorphy is by finding a biholomorphic map between that domain and another domain holomorphy. A biholomorphic map is a map that is holomorphic and has a holomorphic inverse. As we have shown in section 4, unit disk is a domain of holomorphy. Thus, all simply connected domains in $\mathbb{C}$ are domains of holomorphy.

However, an analogue for the Riemann mapping theorem does not exist for several complex variables. Domains of holomorphy has a geometrical property known as pseudoconvexity. The formal description of pseudoconvexity is very technical and we believe that the formal definition is not useful to the audience. Instead, we will present the reader with some domains of holomorphy, and then describe a method for constructing new domains of holomorphy from those building blocks.

We already know that the unit disk $\mathbb{D} \subset \mathbb{C}$ is a domain of holomorphy, and how we can use biholomorphic mappings to find new ones. In higher dimensions, we can construct a domain of holomorphy using a Cartesian product of other domains of holomorphy. As a special case, the Cartesian product of $d$ disks is called a polydisk $D^d(c,r)$ centered on $c$ and with radius $r$ and is defined as

$$D^d(c,r) = \{z \in \mathbb{C}^d \,|\, |z_j - c_j| \leq r_j, j = 1, \cdots, d\}. \tag{107}$$

The boundary of the polydisk $D^d(0,1)$ could be used to encode $[0,1]^d$. To do so, map each real dimension to a complex variable with

$$z_j = e^{i\pi x_j}. \tag{108}$$

The complex exponential is a periodic function that maps the real line to the unit circle. This procedure could be used to complexify datasets like MNIST.

It is possible to find domains of holomorphy that cannot be constructed using a Cartesian product of lower dimensional domains. One such domain is the ball $B(c, r)$ centered on $c$ with radius $r$

$$B(c, r) = \{z \in \mathbb{C}^d \,|\, \|z - c\| \leq r\}. \tag{109}$$

The boundary of the ball could be used to encode correlated dimensions that has a constant magnitude such as one-hot encoded categorical data. The real data could be mapped to a complex variable as the case for polydisks, with the extra step of normalizing $z \in \mathbb{C}^d$ to have $r$ as the magnitude. It goes without saying that a Cartesian product of balls and polydisks is also a domain of holomorphy.

## F    ARE COMPLEX NUMBERS NECESSARY?

One might imagine that we could have described the same framework without ever mentioning the complex numbers. In this section, we discuss how complex analysis helps us in our analysis, and why we think that complex analysis has much more to offer.

First, we know from various papers in the literature of adversarial examples phenomenon that the issue would occur in most applications of machine learning. Thus, if we want to describe the phenomenon, we have to analyse it in a context that is free from the choice of the hypothesis class. In our opinion, an analysis based on learning theory has the best chance of fulfilling this requirement.

However, if we want to apply learning theory, we need to describe the phenomenon in learning theory terms as well. Consequently, we need to come up with definitions that conform to how learning theory defines its objects of study; the language of PAC learnability and uniform convergence.

The first obstacle the we would face is that real differentiable functions are not closed under uniform convergence. In other words, it is possible to find a sequence of real differentiable functions $\{f_n\}_{n=1}^{\infty}$ that is uniformly converging to $f$, and yet $f$ is nowhere differentiable. Consequently, when we are dealing with the output of a learning rule $\mathcal{A}$, we cannot assume that the output is not ill-behaved in its derivatives. We recommend that the reader take a look at the Weierstrass function to get a picture of how ill-behaved the derivatives could become. This is a big issue for our analysis given that most of the definitions in the literature around the context of adversarial examples requires differentiation in one way or the other.

To guarantee that the pointwise limit of a sequence of differentiable functions is differentiable, the sequence needs to be pointwise converging in its value and uniformly converging in its derivatives. According to PAC learnability, to achieve uniform convergence in derivatives, we need to train the derivatives of the hypothesis. But, how would we generate a training set for the gradient of the label "dog" with respect to the pixels in an image? We cannot ask a human to generate the derivatives! On its face, this is likely an impossible feat, and it seems that adversarial examples phenomenon is out of the reach of learning theory.

This is where complex numbers show their true potential. It is known that sequences of complex differentiable (holomorphic) functions are closed under uniform convergence; proposition C.1 is a testament to this fact. In other words, if $\{f_n\}_{n=1}^{\infty}$ is a sequence of holomorphic functions and the sequence is compactly converging to a function $f$, we know that $f$ is holomorphic. As we have stated in the main article, only complex-valued functions of a complex variable can be holomorphic. As a result, moving away from the complex number system to the real number system would be a huge step for some of the proofs in this paper. Nevertheless, while it would be more involved, it is probably possible to find similar theorems for real-valued hypotheses with the help of distributional derivatives.

Apart from the ease of analysis of sequences of holomorphic functions, these functions are also unique in their geometrical properties. In the main article, we freely talk about the angles and lengths in the domain and the range space of a complex-valued hypothesis. We would not be able to do so if it was not for the holomorphicity of the hypothesis. The boundary tilting perspective of Tanay & Griffin (2016) is a good example of the steps needed to be taken for setting up a framework for rigorous study of the phenomenon in a geometrical sense. As demonstrated by Tanay & Griffin (2016), coming up with alternative formal definitions for the real and the imaginary axis in our

proposal is not a walk in the park. Moreover, even when Tanay & Griffin (2016) managed to do so, it proved to be too difficult to apply the geometrical intuition to a nonlinear hypothesis. In contrast, the geometrical interpretability of holomorphic functions enable us to circumvent these problems, and to translate our geometrical intuition to formal mathematical expressions with ease.

In conclusion we argue that while it is possible to recreate our framework without ever mentioning the complex number system, doing so would need even more exotic mathematical objects, and a much more involved discussion.

## G NOTATION AND SYMBOLS

| Symbol | Meaning |
|---|---|
| $\mathcal{A}$ | a learning rule |
| $\mathcal{X}$ | the domain of the hypothesis |
| $\mathcal{H}$ | a hypothesis space |
| $\|\cdot\|_\infty$ | the uniform norm |
| $\partial\mathcal{X}$ | the boundary of the domain $\mathcal{X}$ |
| $\partial\mathcal{H}$ | the class of derivatives of $\mathcal{H}$ |
| $S, S_n$ | a set of training samples with size $n$ |
| d | a distribution |
| $\mathcal{D}$ | a uniform distribution on $\mathcal{X}$ |
| $L^2$ | the space of square-integrable functions |
| $\varphi$ | a feature parameterized by a discreet variable |
| $\sigma$ | a feature parameterized by a continuous variable |
| $\varphi^*$ | a harmonic feature parameterized by a discreet variable |
| $\mathfrak{s}$ | a harmonic feature parameterized by a continuous variable |
| $\mathcal{S}$ | the submanifold of natural samples |
| $\mathcal{C}$ | the submanifold of the decision boundary |
| $\mathcal{O}$ | the space of holomorphic functions |
| $A^2$ | the Bergman space |
| $K_\mathcal{X}$ | the Bergman kernel of $\mathcal{X}$ |
| $f_\mathcal{D}$ | the optimal Bayes classifier |
| $o_\mathcal{D}$ | the holomorphic optimal Bayes classifier |
| $\mathbb{D}$ | the unit disk |
| $\mathbb{T}$ | the unit circle |
| $S_{\partial\mathcal{X}}$ | the Szegő kernel of $\partial\mathcal{X}$ |
| $\|\cdot\|_\mathcal{H}$ | the norm induced by $\mathcal{H}$ |
| $E[h]$ | the Dirichlet energy of $h$ |
| $\nabla$ | the gradient operator |
| $\Delta$ | the Laplacian operator |
| $\mathcal{L}$ | the Lagrangian |
| ReLU | the ramp function |
| $H$ | the Heaviside step function |
| $\{\cdot\}_{n=j}^k$ | a sequence with indices ranging from $j$ to $k$ |

