# OpenReview forum: "An Analytic Framework for Robust Training of Differentiable Hypothesis"
_ICLR.cc/2023/Conference — Submitted to ICLR 2023_

### Official Review · Reviewer_eViJ · 2022-10-22

**Confidence:** 2
**Correctness:** 3
**Technical Novelty And Significance:** 3
**Empirical Novelty And Significance:** 1
**Recommendation:** 5

**Clarity, Quality, Novelty And Reproducibility:**

Novelty: As far as what I know, it looks a good perspective to study adversarial examples.

Reproducibility: This work does not have too many experiments, the code is provided.

Clarify and Quality: I think the presentation of this work needs improvement.
  * There are some latex compilation errors such as "Section ??" in the manuscript.
  * It is better to provide the pseudo-code of the framework in the beginning of Section 5. This will make the readers better aware of the algorithms proposed.
  * Some necessary contexts are needed for some theorems. For example, in Theorem 4.1, the authors should clearly state how $f$ is obtained.
  * Notation is a bit complicated and hard to follow. For example, it is difficult to figure out what is "L with stroke" in Theorem 2.5. What do calligraphic letters $\mathcal{S}$ and $\mathcal{C}$ represent after definition 3.1. Probably, a notation table is necessary in the appendix.


**Strength And Weaknesses:**

Strength:

It is novel to study the phenomenon of adversarial examples from the aspect of differentiable hypothesis, a new aspect as far as what I know.

Weakness and Questions:

1. The settings are a bit restricted: all theorems are based on binary classification. In addition, many theorems are based on support vector classifier (SVC), which is a linear model and much simpler from the ones used in practice.

2. Numerical results are not very strong: a simple MLP model on a single and simple dataset. The attack to evaluate the adversarial accuracy should be stronger than one-step black-box attack (Figure 3). Probably, more experiments are needed.

3. Presentation needs improvement (See the next section)

**Summary Of The Paper:**

This work provides a formal definition for robustness based on learning theoretical terms. Specifically, holomorphicity enables complexity analysis tool to investigate the phenomenon of adversarial examples. In addition, the analysis provides a geometrical interpretation for these phenomena.

**Summary Of The Review:**

In general, I think this paper is not ready for publication because of the concerns above.

As I am not familiar with differentiable hypothesis, I welcome the authors clear my concerns and make me better understand the technical contribution of this paper during the discussion.

---

> ### Author Response · Authors · 2022-11-19
> **Response**
>
> Thank you for your encouraging words. We have fixed the latex compilation errors and fixed the inconsistencies of the notation. We also have added a notation table in Appendix G.
>
> >The settings are a bit restricted: all theorems are based on binary classification. In addition, many theorems are based on support vector classifier (SVC), which is a linear model and much simpler from the ones used in practice.
>
> It is true that we only provide an implementation of the framework for the case of binary classification with SVC. However, we argue that the proposed definition of robustness (def 2.2) is universally true for learning rules (as far as learning theory is concerned) and thm 4.2 could be generalized to all learning rules for binary classification. The analysis for other applications of machine learning is certainly possible and we also have tried to find a generalization of thm 4.2 for all differentiable loss functions. However, our impression is that if such a theorem exists, the proof is not trivial and would need its own text.
>
> > Numerical results are not very strong: a simple MLP model on a single and simple dataset.
>
> While more experiments never hurts, we cannot go beyond small-scale problems right now. The main obstacle is the curse of dimentionality in computing the Dirichlet energy of a hypothesis. In case of polynomials, the curse of dimentionality shows itself through the combinatorial explosion of the count of the polynomial bases in high dimensions. In case of ANNs, it shows itself through NP-hardness of computing the volume of an arbitrary polytope (try computing the Dirichlet energy of a ReLU activated single-layer perceptron network). The proposed PDEs are our solution for this problem but solving the PDEs need its own space.
>
> > It is better to provide the pseudo-code of the framework in the beginning of Section 5. This will make the readers better aware of the algorithms proposed.
>
> We have added a summary of the framework from a practitioner's perspective in appendix A.
>
> > Some necessary contexts are needed for some theorems. For example, in Theorem 4.1, the authors should clearly state how $f$ is obtained.
>
> This is a mistake from our part, $f$ needs to be replaced by $h$. Hopefully, the overall changes have made the reasoning behind the proposed definitions and theorems more tangible.

---

### Official Review · Reviewer_2yAx · 2022-10-24

**Confidence:** 3
**Correctness:** 4
**Technical Novelty And Significance:** 4
**Empirical Novelty And Significance:** 3
**Recommendation:** 8

**Clarity, Quality, Novelty And Reproducibility:**

The authors’ contributions look novel, and the writing is straightforward and understandable. There might be a few reproducibility issues.


**Strength And Weaknesses:**

[[Strengths]]
1. The paper provides precious theoretical insights into robust classification.
2. About implementing the authors’ learning rule (Theorem 4.2), the paper proposes a practical method based on a well-understood PDE problem. This reduction to the PDE problem would open up a new research direction.

[[Weaknesses]]
1. The paper has a few LaTeX issues. First, there are some missing references, e.g., ‘section ??’ in Section 5, ‘lemma ??’ in Appendix C.4, and ‘definition ??’ in Appendix C.8. Since those missing references appear in the main body (or even in the proof), the authors should fix these missing references. Also, please fix the Appendix sectioning so that the material starts from Appendix A. (Currently, Appendix A is just an empty section with the title ‘Appendix’)
2. About the experimental details, the authors mentioned that “The interested reader can find the details of the experiment in the supplementary materials.”, but the supplementary materials only contain iPython notebooks that the reader should run. This is submitted as an academic paper, and general readers (even those who are interested in experimental details) only read the written materials. Therefore, this raises a reproducibility issue. (the descriptions in the paper are not enough to reproduce the result.) I suggest the authors 1. write an additional section (in the Appendix) about experimental details (e.g., network structure, hyperparameters, etc.) and 2. put your code on a code repository and add the link to the repository in the camera-ready version. (Because this is a blind review, you should not add the repository link during the review process.)


**Summary Of The Paper:**

This paper provides an analytical framework to understand adversarial example problems in a more formal way. Several hypothesis classes are discussed, especially $L^2$ (square-integrable functions) and $A^2$ (Bergman space, a subspace of holomorphic square-integrable functions). For their framework, the authors extended the space of classifiers so that it contains complex-valued functions as classifiers. Their main statement (Theorem 4.2) states that a learning rule minimizing Dirichlet energy would induce a robust classifier. A practical discussion on how to implement the main idea and an experimental exploration are provided.


**Summary Of The Review:**

I consider the paper novel and valuable work. This paper contains novel insights into training robust classifiers. Especially the concept of complex-valued function as a classifier can provide further insights as it simultaneously encodes two geometric objects (the data manifold and the decision boundary). Though I could not check the whole proof, the suggested learning rule looks novel and interesting. The author also demonstrated that the proposed idea is also implementable and can be reduced to solving a PDE problem.
Regarding some negative factors, the paper contains some LaTeX problems, including missing references, and lacks details about the experimental setup, raising a reproducibility issue. However, I believe these are amendable issues, and I’m willing to raise the score after they are fixed.

---

> ### Author Response · Authors · 2022-11-19
> **Response**
>
> Thank you for your kind words. We have fixed the latex compilation errors and filled Appendix A with a summary of the framework. We also have added Appendix D and went into details about the experiments in the paper to help with the reproducibility issue. We will add a link to a repository for the camera-ready version per your suggestion.

---

> > ### Comment · Reviewer_2yAx · 2022-11-26
> > **Thank you for the response**
> >
> > Thank you for the update. I believe that Appendix A and Appendix D can cover my concern of reproducibility. Because I don't see any other concern and the paper looks to have enough achievement, I adjust my score accordingly.

---

### Official Review · Reviewer_od7j · 2022-10-24

**Confidence:** 2
**Correctness:** 3
**Technical Novelty And Significance:** 3
**Empirical Novelty And Significance:** 3
**Recommendation:** 5

**Clarity, Quality, Novelty And Reproducibility:**

Clarity: This paper is not very well written.
Quality: This paper is technically sound.
Novelty: The novelty of this paper is high.

**Details Of Ethics Concerns:**

No.

**Strength And Weaknesses:**

Strength: The idea of using calculus of variations and differential equations to tackle the robust training of neural network models is novel and interesting. The theoretical results seem solid.

Weaknesses: 1. This paper is not well written since there are many typos. For example, see "Section ??" on Page 8, "Lemma ??" on Page 13, and "Definition ??" on Page 15.
2. Many theoretical results are about SVC. More theoretical results on more complicated models should be included.

**Summary Of The Paper:**

In this paper, the authors introduced a general framework for training robust models. The idea of using calculus of variations and differential equations to tackle the robust training of neural network models is novel and interesting. Some theoretical analysis is given.

**Summary Of The Review:**

In summary, this is a solid paper with novel ideas. The idea of using calculus of variations and differential equations to tackle the robust training of neural network models is novel and interesting. However, This paper is not very well written. Also, more theoretical results on more complicated models should be included.

---

> ### Author Response · Authors · 2022-11-18
> **Response**
>
> Thank you for the encouraging words. We have updated the paper to address some of the issues in your review. We are hopeful that the reviewer would increase the score after discussion.
>
> > This paper is not well written since there are many typos.
>
> We have fixed the latex compilation errors and notation inconsistencies.
>
> > Many theoretical results are about SVC. More theoretical results on more complicated models should be included.
>
> The main result of the paper is thm 4.2 which is universally applicable to any learning rule as long as it could be shown that def 2.2 is satisfied. For classifiers, irrespective of the learning rule, minimizers of Dirichlet energy are robust. Other applications of machine learning need to be analyzed separately, but the skeleton of the proves are the same as long as differentiability could be assumed.

---

### Official Review · Reviewer_vhPa · 2022-10-28

**Confidence:** 3
**Clarity, Quality, Novelty And Reproducibility:** I have expressed my concerns on clari…
**Correctness:** 2
**Technical Novelty And Significance:** 2
**Empirical Novelty And Significance:** 2
**Recommendation:** 3

**Strength And Weaknesses:**

Strength:
The paper tries to provide a general framework for robust learning and the goal is plausible.

Weaknesses:
The writing is weak and as a result I can only barely judge the other aspects of the paper.
Aside from the obvious LaTex compiling error, the paper spends quite a few pages to include new definitions, which are only explained in later sections. When reading the paper following each section, the reader may get confused about why this definition is included -- very oftenly the definition just appear without any motivating sentences. For example, there is no explanation for why SVC is needed before Definition 2.4 and why complex-value classifier is needed before Definition 3.1.
The writing also fails to clearly identify Assumptions that should be made for this analytical framework, e.g., "If we find some sample x that does not perplex h but is not correctly labeled by h, then we assume that H is agnostic to the pattern f x." appears in a paragraph, but I think this should be an assumption that needs to get highlighted.
Also in terms of clarity, when the paper mentions "integral representation is more adequate for analysis when there is a continuum of features to choose from, e.g. ANN" -- the paper does not define what 'feature' is in ANN so one can only guess what the meaning is.
There are clarity issues in the main results section (Section 4) as well. Theorem should be self-explanatory as much as possible, whereas in Theorem 4.1, it is unclear what does f mean and there is no short proof provided (if one dives deeper, it points to Appendix C.7 then Proposition C.2 which seems to be from another paper.).


**Summary Of The Paper:**

The paper proposes a geometric and analytic modeling perspective for robust training. They first propose a formalization of robustness in learning theoretic term, then give a geometrical description of the phenomenon for simple classifiers. Experiments are conducted on synthetic and real-world data to verify their idea.

**Summary Of The Review:**

I would vote for reject since the writing does not meet the bar. Even though the paper may contain interesting ideas, it probably should go through another review round after a major revision.

---

> ### Author Response · Authors · 2022-11-18
> **Response**
>
> Thank you for your honest and to the point review. We have fixed the latex compilation errors and notation inconsistency. We made some changes to the paper to address the issues. Hopefully the changes would compel the reviewer to increase the score.
>
> >  the paper spends quite a few pages to include new definitions, which are only explained in later sections.
>
> We have added appendix A that aims to be something like a TL;DR summary of the framework. Hopefully, this section would motivate the reader to challenge herself to go through the definitions and the proves of the theorems. The framework is about proving uniform convergence (def 2.2) based on Cauchy's criterion and the paper tries to both describe and give a demo of the process. In our defense, we have no choice but to go through some definitions first because we do not believe that some of the basic definitions could be considered as common knowledge.
>
> > The writing also fails to clearly identify Assumptions that should be made for this analytical framework, e.g., "If we find some sample x that does not perplex h but is not correctly labeled by h, then we assume that H is agnostic to the pattern f x." appears in a paragraph, but I think this should be an assumption that needs to get highlighted.
>
> Section 2 is about the first assumptions of the framework. The analytical part of the framework is not an assumption, complex analysis is a tool for proving thm. 4.2. After establishing thm. 4.2, the requirement for complex differentiability reduces to differentiability. Nevertheless, complex differentiable hypotheses will remain to be an important class because almost all activation functions that are common in ANNs are either meromorphic (e.g. tanh, atan, ...) or could be well approximated with a meromorphic function. The relation between meromorphic functions and holomorphic functions is the same as the relation between rational functions and polynomials. So analyticity is not a requirement. The reason for the sentence in the paragraph is to highlight that if adversarial training of an adversarial example did not make the hypothesis robust, then a solution could not be sought using our framework. We have added a sentence to the paragraph to make this assumption more explicit.
>
> > the paper does not define what 'feature' is in ANN so one can only guess what the meaning is.
>
> We have updated the text to point to a perceptron.
>
> >  in Theorem 4.1, it is unclear what does f mean.
>
> "f" should be changed to "h", this is a typo that is fixed now. The root of this problem is a last day notation change that introduced latex compilation errors and notation inconsistencies.

---

### Author Response · Authors · 2022-11-19
**Revision details**

- change the activation function used in the integral representation of the hypothesis in figure 2 from ReLU to the Heaviside step function $H$. Apart from a better computational stability, We think that $H$ is theoretically more significant than ReLU. The relevant part in the text is updated to reflect the change.

- add Appendix A per recommendation of the reviewers. The section aims to be a summary of the proposed framework.

- add Appendix D per recommendation of the reviewers. The section aims to provide the necessary details for reproducing the experiments in the paper.

- add Appendix G per recommendation of the reviewers. The section contains a notation table

- fix latex compilation errors and notation inconsistencies

- add or change a few sentences to make some assumptions and details explicit in the text

---

### Decision · Program_Chairs · 2023-01-20

**Decision:**

Reject

**Justification For Why Not Higher Score:**

N/A

**Justification For Why Not Lower Score:**

N/A

**Metareview: Summary, Strengths And Weaknesses:**

The paper proposes a geometric and analytic modeling perspective for robust training. The idea is novel. But all the reviewers find the paper is not ready for publish. For example, there are many latex errors, many definitions and experimental details are missing. The paper should be carefully revised before publish.